# An efficient real-time signal processing method for satellite laser ranging

Jinyu Wang[1], Wei Zhu[1,2]*, Weiming Gong[1,2]

1 Institute of Seismology, China Earthquake Administration, Wuhan, Hubei, China, 2 Hubei Key Laboratory of Earthquake Early Warning, Hubei Earthquake Agency, Wuhan, Hubei, China

* 15972138235@163.com

**Data Availability Statement:** All relevant data are within the paper and its Supporting information files.

**Funding:** The authors report no sources of funding.

## Abstract

In recent years, various real-time processing methods have been developed for Satellite Laser Ranging (SLR) data. However, the recognition rate of the single-stage Graz filtering algorithm for high-orbit satellites is less than 1%, and traditional two-stage filtering algorithms, such as polynomial fitting and iterative filtering techniques, exhibit high false and missed detection rates. These issues compromise the accuracy of laser positioning and real-time adjustments during observations. To address these problems, we propose a new, efficient real-time SLR data processing method. This algorithm combines single-stage filtering with a histogram-based approach and incorporates polynomial fitting to establish a predictive model. This offers the advantage of fast and efficient real-time signal recognition. The experimental results demonstrate that the proposed algorithm compensates for the limitations of single-stage filtering algorithms and performs better than traditional two-stage filtering algorithms in identifying medium- and high-orbit satellite signals. The false detection rate was reduced to below 15%, while achieving faster computation speeds. This method is convenience for researchers in their observations and offers new insights and directions for further research and application in the real-time identification of satellite laser ranging echo signals.

## Introduction

Satellite laser ranging (SLR) is a high-precision geodetic method used in many fields, such as geodynamics, geodesy, and astronomy [1–4]. SLR obtains the distance between the Earth and the satellite by transmitting laser pulses that accurately measure the time from the ground observation point to the satellite equipped with a corner reflector [5–7]. Currently, with the continuous advancement of kHz satellite laser technology, the volume of echo data has significantly increased [8–11]. However, due to the influence of sky background noise and system noise, a substantial amount of noise is present in the echo signals [12–14]. This issue is further exacerbated in observations of high-orbit satellites and space debris, where the echo signals are notably weaker [15–17]. During observations, operators must adjust the laser's position based on whether valid signals are present in the echo data to ensure the laser accurately targets the

**Competing interests:** The authors have declared that no competing interests exist.

satellite. Therefore, effective data processing is crucial for optimizing laser ranging observations and ensuring precise satellite tracking.

The manual screen recognition method relies on a human-computer interaction interface, where observers manually filter valid signals based on experience. While it offers strong adaptability and effectively filters noise, it involves extensive manual intervention, has low automation, and results in reduced efficiency. [18–20]. The Poisson filtering algorithm segments data over short time intervals using tilted grids. When the number of echo points within a grid exceeds a predefined threshold and follows a Poisson process, the region is determined to contain valid echo data. This method performs well in practice. However, its detection results are significantly influenced by Poisson statistical filtering, and it lacks strong adaptability in data processing [21–23]. To improve the automation of processing data, various histogram methods have been proposed to determine whether the echoes in the box are valid or not by setting the size of the distance window, the moving interval, and the threshold according to the statistical characteristics of the SLR echoes [24]. Graz fast echo recognition algorithm [25, 26], is one of the typical algorithms, the algorithm compares the current point with the previous 1,000 points by setting a threshold to filter valid signals that meet the criteria. While fast and effective in low-orbit satellite signal processing, its recognition rate for far-Earth satellites is below 1%.

When the algorithm filtering effect is poor, and there are a large number of noises are mislabeled, the filtering effect can be improved by increasing the comparison point $n$ and the set threshold, but the increase in the value of comparison point $n$ increases the amount of computation of the computer, and the increase in the threshold in the case of low echo rate, the rate of leakage detection will increase [27]. Therefore, the method of multiple iterations is used to improve the filtering effect, Qin Si [28] proposed a two-filter algorithm strategy, where a second Graz filter is applied after the first, comparing the current point with the previous 300 points. If the threshold is met, the signal is considered valid. The secondary filter adds a small amount of computation but does not affect the real-time nature of the signal recognition and improves the filtering effect, the disadvantage is that it performs poorly in the processing of the satellite data with weak echo signals and has a high rate of miss detection. T.M. Ma combined primary filtering with polynomial fitting, using a first-stage Graz filter to perform linear fitting on the points that meet the filtering criteria. After excluding outliers, a new curve is fitted, and the current point is evaluated by calculating its residual against the new curve. If the residual is below a set threshold, the signal is considered valid. Compared to primary filtering, this approach enhances noise exclusion and improves the recognition of weak signals, However, the false recognition rate is high [29]. Literature [30] proposed a deep learning data processing method, using the residual maps of manually identified signals and those recognized by the GRAZ algorithm as training targets for a deep neural network to identify valid signals. Additionally, an autoencoder was employed to remove noise, which increased the level of automation compared to manual recognition. However, this method has not been applied to range data with significant prediction deviations and, like the GRAZ algorithm, it is unable to identify echo signals from high-orbit satellites. Zequn Lv [31] employs image denoising technology to enhance data processing efficiency within SLR systems. This method identifies valid signal points by binarizing the image and using median filtering for noise removal. The approach significantly improves the accuracy of data recognition; however, its application is limited to the preprocessing stage and does not support real-time recognition of valid signals.

In summary, while many of the aforementioned methods demonstrate good performance, they fall short in achieving real-time signal processing. Techniques such as Poisson filtering and manual screen recognition effectively handle noise but are either too slow or require significant manual intervention. Histogram-based methods, including the Graz algorithm, show

excellent results for low-orbit satellites; however their recognition rate for far-Earth satellites is inadequate. Similarly, although polynomial fitting and the two-stage Graz filtering algorithm can process signals in real time, enhancing noise exclusion and weak signal recognition, they exhibit high false and missed detection rates. This hinders the ability of the operators to adjust the laser position in a timely manner, making these methods less effective for practical observation tasks.

To address the challenge of weak echo signals and high noise levels in satellite laser ranging for medium- and high-orbit satellites, which hinders real-time and accurate signal identification, we propose a novel and efficient real-time signal processing method. This method employs a multi-stage filtering strategy, inheriting the speed of the Graz filtering algorithm while integrating a histogram-based secondary filtering approach to improve the recognition rate for medium- and high-orbit satellites. Additionally, we apply the least squares method to fit strong echo signals into a predictive straight line, thereby establishing a predictive model that mitigates the high false detection rate associated with traditional two-stage filtering algorithms. The proposed algorithm was validated using real-world observation data, demonstrating its capability to accurately and reliably identify weak signals in SLR in real-time. The experimental results confirm the effectiveness and robustness of the algorithm.

To clearly demonstrate the performance differences between existing methods and the proposed method, the Table 1 below compares various processing methods and highlights the advantages and disadvantages of each, including our new filtering method.

## Theoretical foundation and limitations of the secondary filtering algorithm

### Primary filtering algorithm

The Graz algorithm, a typical histogram processing method, is known for its speed and efficiency, making it effective for quickly identifying satellite data in medium- to low- Earth orbits. During the short observation periods, the echo signal O-C residuals exhibited minimal variation, with most valid signal points clustered around a straight line, as shown in Fig 1. Here, the O-C residual represents the difference between actual and predicted values. Fig 2 provides a partial magnification of Fig 1, demonstrating that the distribution of valid signals within a short time frame approximates a straight line. Upon zooming in, it is evident that the O-C residuals of the valid points fall within a specific range. Based on the characteristics of the

**Table 1. Comparison of different processing methods.**

| Method | Advantages | Disadvantages | Performance Metrics |
|---|---|---|---|
| Manual Screen Recognition | High adaptability, effective noise removal | High manual involvement, low automation, low efficiency | High human involvement, slow processing |
| Poisson Filtering | Good at noise removal, simple implementation | Sensitive to Poisson filtering assumptions, low adaptability | Only data preprocessing |
| Histogram-based Methods | Fast processing, effective for low-orbit satellites | Poor performance for far-Earth satellites | Excellent for low-orbit, poor for high-orbit satellites |
| Two Graz and Graz +polynomial | Improved noise rejection, real-time processing | High false detection and missed detection rate | Certain inconveniences in practical applications |
| Deep Learning | Automates signal recognition, high adaptability | Requires large datasets, high computational cost | The real-time signal processing cannot be applied |
| Image Denoising | Enhances data quality, effective for preprocessing | Not suitable for real-time signal recognition | High preprocessing accuracy, no real-time capability |
| New Algorithm | Accurately identify valid signals, real-time processing | The adaptability is limited | Real-time and accurate identification of weak signals |

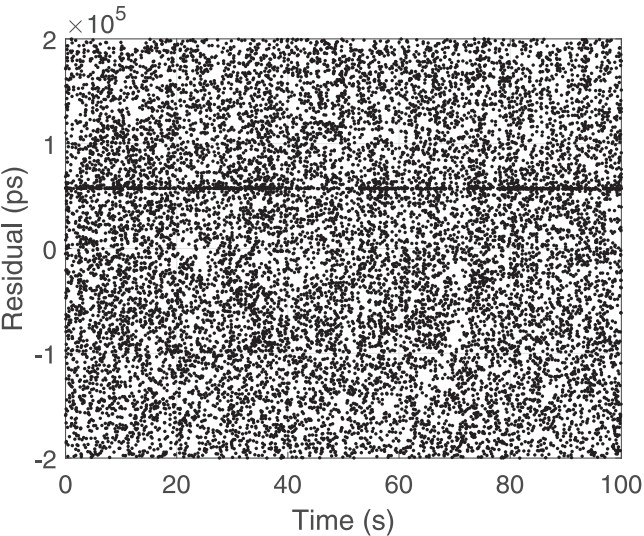

**Fig 1. Glonass107 satellite O-C residual map.**

valid signal distribution, the residual of the current point and those of the previously obtained points are subtracted from each other. The filtering expression is as follows:

$$|\Delta R_i - \Delta R_{in}| < \delta \qquad (1)$$

where $\Delta R_i$ is the O-C value of the current point, $\Delta R_{in}$ is the O-C value of the n-th point prior to the current point, and $\delta$ is the comparison threshold, which is generally set based on observation accuracy and prediction error.

The specific filtering strategy is as follows: The absolute difference between the O-C value of the current data point and the O-C values of the previous n data points (for example, n = 1000) is computed and compared with the threshold $\delta$ (for example, $\delta$ = 1000 ps). If the

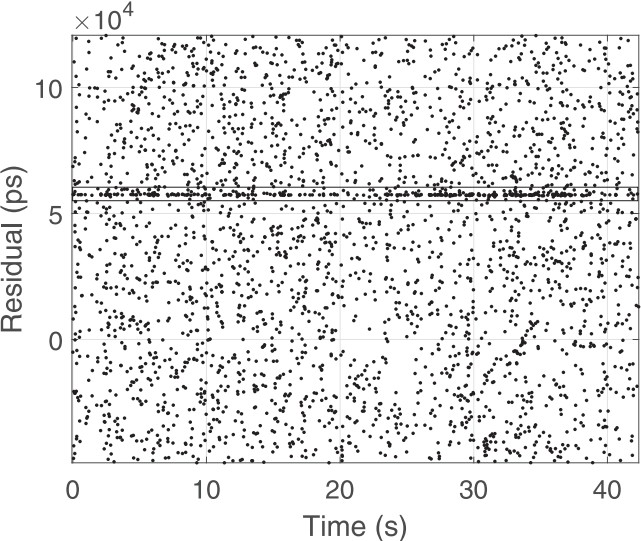

**Fig 2. The localized magnification of the O-C residuals for the Glonass107 satellite.**

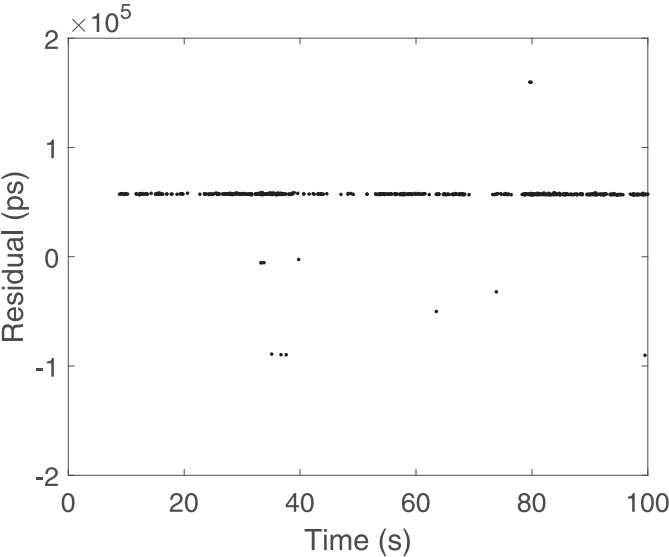

**Fig 3. Two Graz recognition result.**

number of points satisfying the condition of being less than the threshold is greater than or equal to m (for example, m = 3), the current point is marked as a valid echo.

When the filtering performance of the algorithm is poor and a significant number of noise points are incorrectly marked, increasing the values of n and m can improve the filtering effectiveness. However, increasing n increases the computational load, and increasing m under low echo rates may increase the miss detection rate. For medium-orbit and distant satellites, the echo signals are often too weak to meet the set threshold conditions. If the threshold is relaxed, many noise signals are incorrectly recognized, resulting in a recognition rate of less than 1% for distant satellites. Therefore, a multiple iteration approach is employed to enhance the filtering performance.

## Secondary filtering algorithm

When the performance of the Graz algorithm is insufficient, a two-stage Graz filtering strategy is employed. This involves the application of a second filtering process in addition to the initial filtering process. During the second filtering stage, n was set to 300, m was set to 3, and threshold $\delta$ was set to 500 ps. If first-stage filtering is satisfied, the O-C value of the current point is compared with the O-C values of the previous 300 data points. If at least three of these comparisons yielded results of less than 500 ps, the current point was considered a valid echo.

The addition of the secondary Filtering Algorithm increases the computational effort slightly but does not affect the real-time signal identification. This enhanced the filtering performance. However, a notable drawback is its poor performance with weak echo signals in satellite data, which leads to a higher false detection rate. Because the Graz filtering algorithm compares the current point with the previous n points, it does not process the data if the index of current point is less than n, resulting in a certain delay in signal recognition during the initial stage. When applying the two-stage Graz filtering algorithm, this delay becomes even more pronounced, with the system only identifying valid signals four to five seconds after the laser has hit the satellite. The Figs 1 and 3 shows the raw measurement data from the GLONASS satellite and the results after applying the two-stage filtering process.

## Graz+ polynomial recognition algorithm

Similar to the two Graz filtering algorithm, this method applies an additional filtering stage based on the initial Graz filtering. In the second filtering stage, n was set to 30. The standard deviation of the O-C values from these 30 points was used to eliminate data points with large errors. The O-C values of the remaining data points were used to calculate the fitting residual, denoted by $\sigma$. The fitting residual $\sigma'$ was then computed based on the fitted line and the current O-C value. If the fitting residual of the current echo signal satisfies $\sigma' < 2.5\ \sigma$, the current echo signal is considered valid.

By combining initial filtering with polynomial fitting, this method enhances the capability of eliminating noise and improves the detection of weak signals compared to single-stage filtering. The Graz + Polynomial Recognition Algorithm demonstrates an improved filtering performance, resolving the delay issue of the two-stage Graz filtering algorithm by identifying valid signals earlier and reducing the number of missed signals. However, it also exhibits a higher false detection rate, with many noise signals incorrectly labeled. Fig 1 shows the raw measurement data from the GLONASS 107 satellite, and the results after filtering are presented in Fig 4.

## New filtering algorithm

To address the limitations of the single-stage Graz filtering algorithm and traditional two-stage filtering algorithms while retaining their advantages, this study leveraged the dense distribution of valid signals in satellite laser ranging echoes and their tendency to approximate a straight line over short time intervals. Based on the Graz filtering algorithm, we apply an additional histogram-based filtering method to reduce the false detection rate. The recognized signals are then fitted into a straight line using the least squares method, with the aim of identifying valid signal points under low echo rate conditions and reducing the missed detection rate.

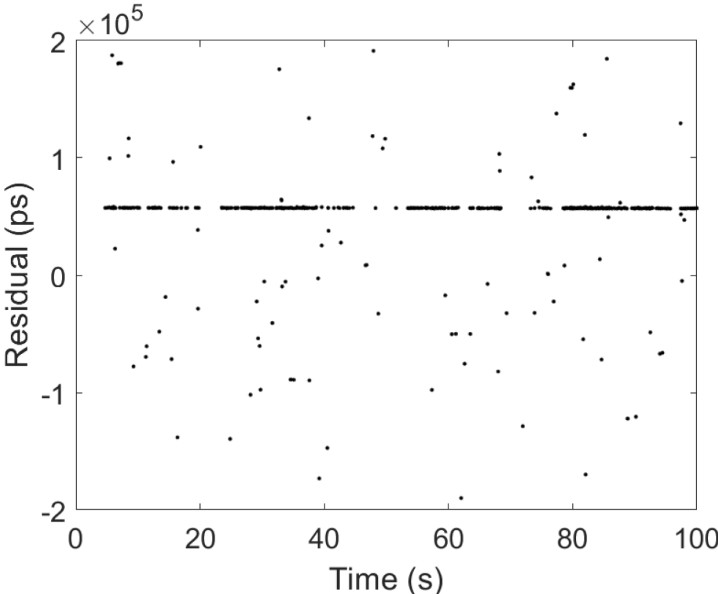

**Fig 4. Graz+ polynomial recognition result.**

When the Graz filtering algorithm is satisfied, secondary filtering is started, a large square grid is generated in the current observation area, the signal processing area is inside the large square grid, and the distance of each translation of the large square grid in the positive direction of the x-axis is 0.5s. A small square grid $\alpha$ with a width equal to that of the large square grid and a height of q (set to 1000 ps) is moved upward from the lowest end of the large square grid by moving upward by 0.5q (500 ps) each time. Record the number of signal points at each position during the moving process, and find the position of the small square grid $\alpha$ with the highest number of signal points; the number of signal points in this position is recorded as m. When m is greater than or equal to $\theta$ (set to 6), it is regarded as a reliable and valid signal, and look back at the recent 20s, and fits the reliable and valid signal points into a prediction straight line. The predicted straight line can be used to judge the current situation where the number of points in the small square grid $\alpha$ is less than $\theta$. Signal points within 250 ps of the prediction straight line were labeled as valid echoes. The prediction straight line can also correct the case where m is greater than or equal to $\theta$. Signal points in the small square grid $\alpha$ that are too far away from the prediction straight line are regarded as noise and deleted.

1. Initialization: After a data point passes the initial Graz filter, the position of the first filtered point is recorded. This position was used to generate a large grid with a width of 2s and height of 400 ns, with the grid height ranging from ±200 ns. The grid is shifted 0.5s in the positive x-direction every 0.5s.

2. Grid Analysis: As new data points are generated, if the x-coordinate of the next data point falls to the right of the large grid, a small grid with a width of 2s and height of 1000 ps is moved upwards from the bottom of the large grid, with each movement being 500 ps. Record the position of each movement of the small grid and the number of data points within it. Identify the position of the small grid with the highest number of signal points and the corresponding number of data points m.

3. Signal Point Identification: If the number of data points m within the small grid is greater than or equal to 6, the data points within that grid are considered reliable valid signal points. If no prediction line is available, the signal is directly recognized as a valid echo.

4. Prediction Line Fitting: If reliable valid signals exist in the last 20 seconds, fit these signals into a prediction line.

5. Validation Against Prediction Line: If a prediction line is present, calculate the distance of data points within the corresponding grid from the prediction line. Data points within a distance of no more than 0.25h (250 ps) from the prediction line are considered valid echoes.

6. Iteration: Repeat steps 2 to 5 until observation stops.

The flowchart for the online echo signal extraction algorithm is shown in Fig 5.

Refer to Fig 6. The width of the large square grid is set to 2s. If the moving distance of the large square grid is equal to its width, the algorithm will mark the generated data points every 2s, which is significantly delayed and is not conducive to the timely adjustment of the laser position of the observers, the width of the square grid is too large, then the algorithm calculates the time to increase the width of the square grid is too small, and the algorithm has a poor filtering effect. Therefore, the moving distance is 0.5s, and the width is set to 2s so that the algorithm can mark the signal point every 0.5s, enhance the timeliness of the algorithm, and take into account the filtering effect of the algorithm.

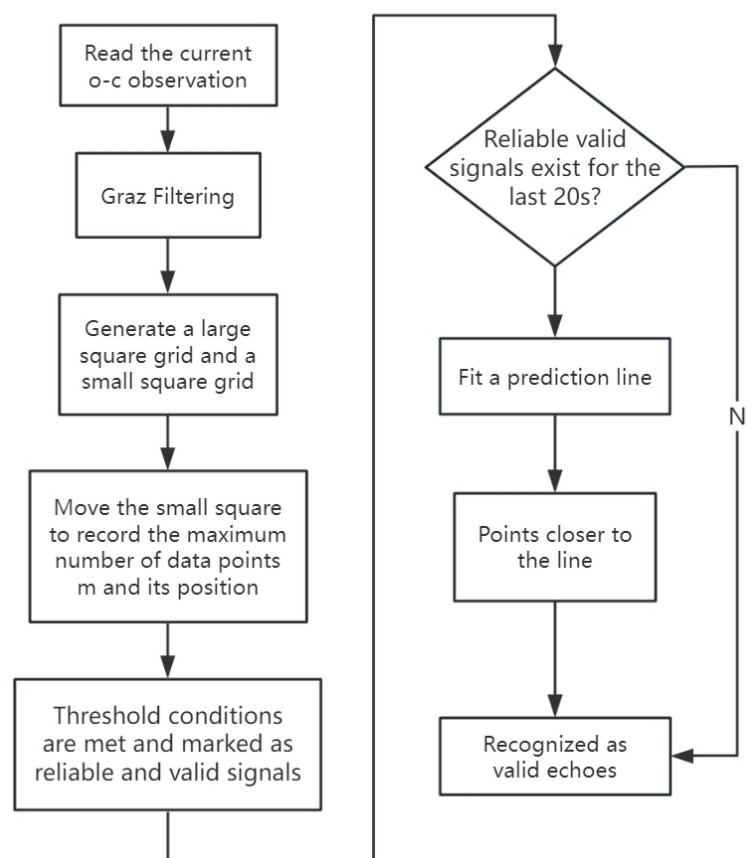

**Fig 5. Flowchart of the new algorithm.**

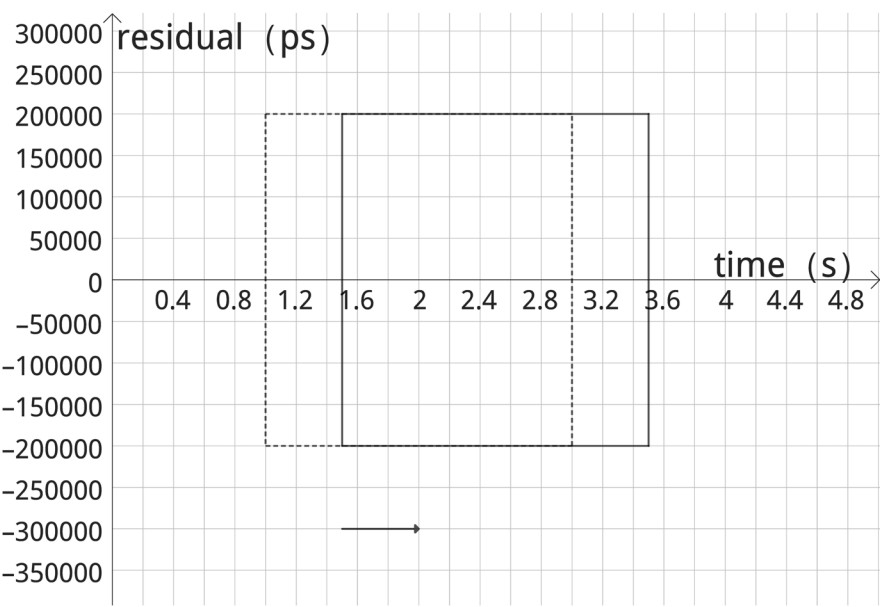

**Fig 6. Large square grid moving map.**

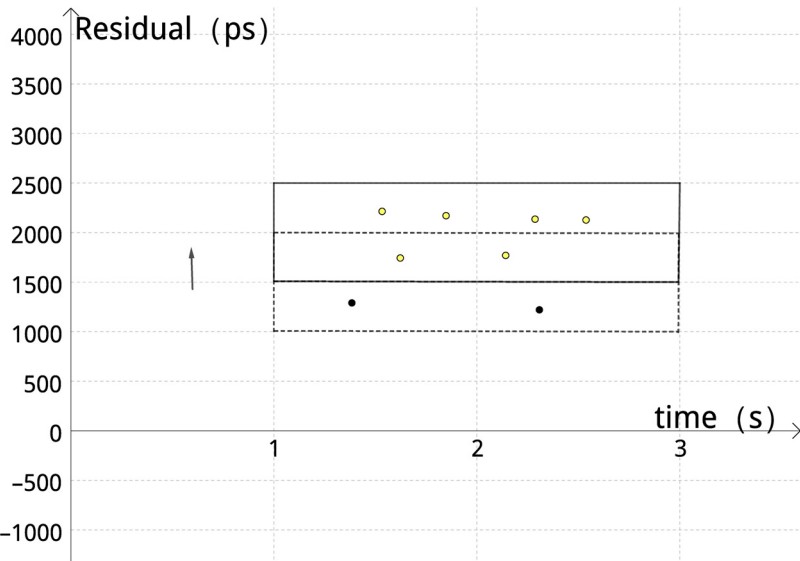

**Fig 7. Small square grid moving map.**

In the process of moving the small square grid, to avoid the signal point falling into the boundary of the square caused by the ambiguity of the subsequent algorithm processing, the moving distance of the small square is set to half of its height, using the characteristics of the effective signal in a short period can be seen as a straight line approximately, and each moment ultimately only need to retain the data in a small square, rather than all the data, to make the algorithm more rapid, and if the number of data m in the small square is more than the set threshold $\theta$ then it will be marked as a reliable and valid signal point, for example, the yellow signal points in Fig 7 are reliable and valid signal points, and the black points are regarded as noise.

To address the issue of signal points in the small square grid falling below the set threshold $\theta$ (e.g., $\theta$ is taken as 6) and prone to missed detection, a predictive line is fitted using the least squares method by examining the most recently marked reliable valid signal points. Signal points within the distance range of the straight line ±250 ps were then labeled as valid signals, as illustrated in Fig 8. In the figure, the yellow signal points are labeled as valid echoes, while the black points are considered noise and are to be removed from the line.be removed from the line.

## Results

To test the recognition effect of the algorithm, data obtained from the ranging experiments at the Xinjiang Nanshan Station of the Seismological Research Institute of the China Earthquake Administration were utilized for processing. The laser frequency of the satellite laser ranging mobile station was 1 kHz, the software platform for algorithm calculation was MATLAB under the Win11 system, the memory of the computer was 16 GB, and the CPU was AMD Riptide 7 7735H with a main frequency of 3.2 GHz. The parameters of the processed observational targets, such as time and arc length, are listed Table 2 below. The measured data O-C values of three satellites with different orbital altitudes are processed with two traditional secondary filtering algorithms and the echo signal recognition algorithm proposed in this paper.

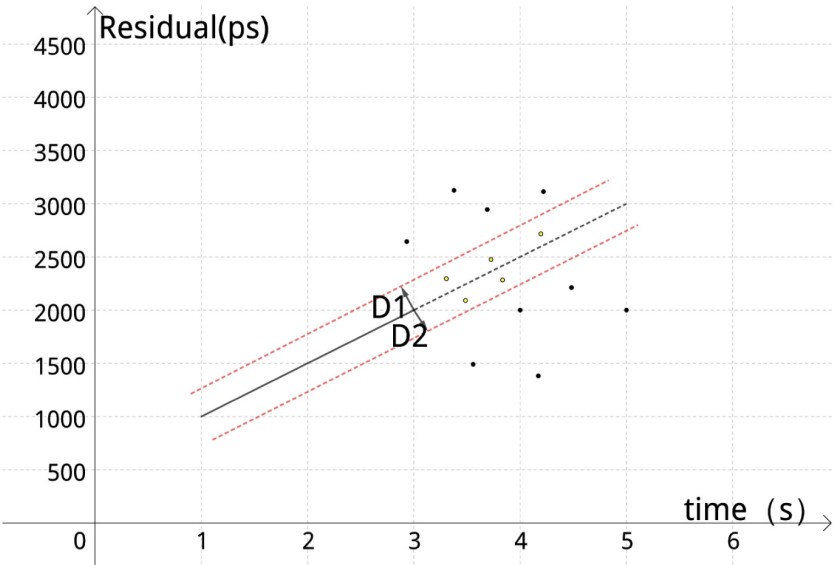

**Fig 8. Predicted straight-line display.**

Each satellite records data for 100 s, with 100,000 data points and results of the processing of the three algorithms are shown in Figs 9–11.

From the above Figs 9–11, for the echo data of different orbital satellites, the recognition algorithm proposed in this paper has a fast computational speed, can effectively recognize the weak echo signals online, can still accurately recognize them even if there is a signal discontinuity, recognizes a rapid response, recognizes the effective echoes earlier than the two traditional algorithms, does not incorrectly recognize the apparent anomalies in the recognition process, and has significant filtering effects. A prediction line is generated based on the latest identified valid signal points such that even if the valid signal is feeble during observation and the number of signal points is below the filter threshold, the signal can be accurately identified because of the prediction line. To judge the performance of the algorithms more rigorously, compare with manual screening method, the algorithms are tested by setting two parameters, namely, $F_r$ (the false detection rate) and $L_r$ (the leakage rate), and the results are as follows:

$$F_r = \frac{\text{Algorithmic misidentification points}}{\text{Total Algorithm Detection Points}} \times 100\% \tag{2}$$

$$L_r = \frac{\text{Points not recognized by the algorithm}}{\text{Total Algorithm Detection Points}} \times 100\% \tag{3}$$

**Table 2. Satellite laser ranging experiment results.**

| Objectives | Etalon1 | Etalon2 | Lares2 |
|---|---|---|---|
| Time | 2023–09–06 | 2023–09–11 | 2023–0816 |
| Observation arc length [s] | 769 | 568 | 246 |
| Perigee[km] | 20505 | 20068 | 6699 |
| Apsis [km] | 20591 | 20251 | 6649 |
| Return Points | 139226 | 98486 | 39132 |

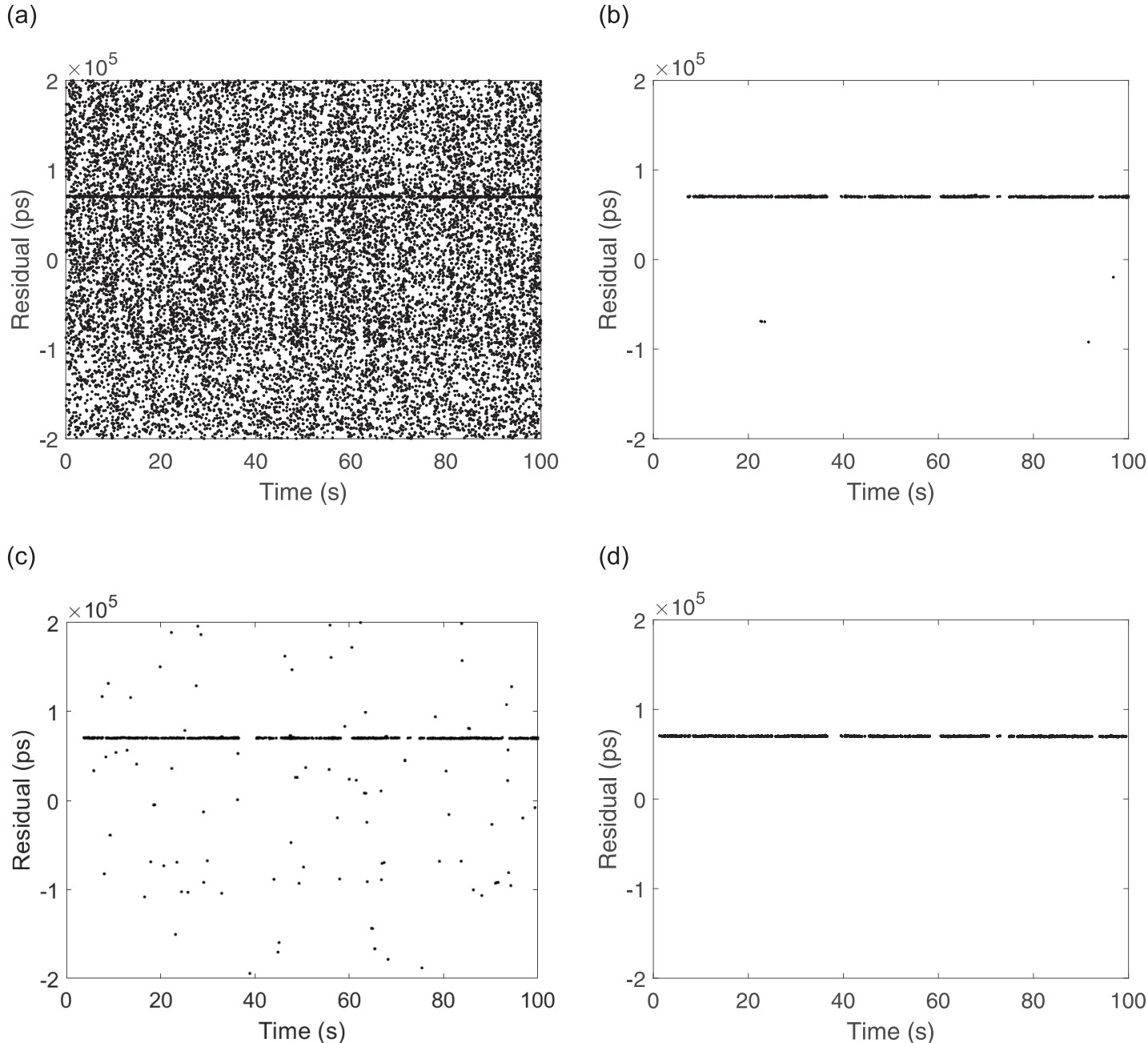

**Fig 9. Etalon1 observations and algorithmic processing results.** (a) Etalon1 raw observation data, (b) Two Graz, (c) Graz+polynomial, (d) New Algorithm.

The lower the values of the two parameters of false detection rate and leakage rate, the better the performance of the algorithm; as can be seen from Fig 12 and Table 3, the three algorithms in the leakage rate of the recognition effect of the difference is not large, but in the false detection rate of the algorithm proposed in this paper is significantly lower than the other two algorithms, the signal processing effect is better.

The signal recognition algorithm proposed in this paper has been verified to prove that it has a low false detection rate and leakage rate, to verify its timeliness, the raw data collected

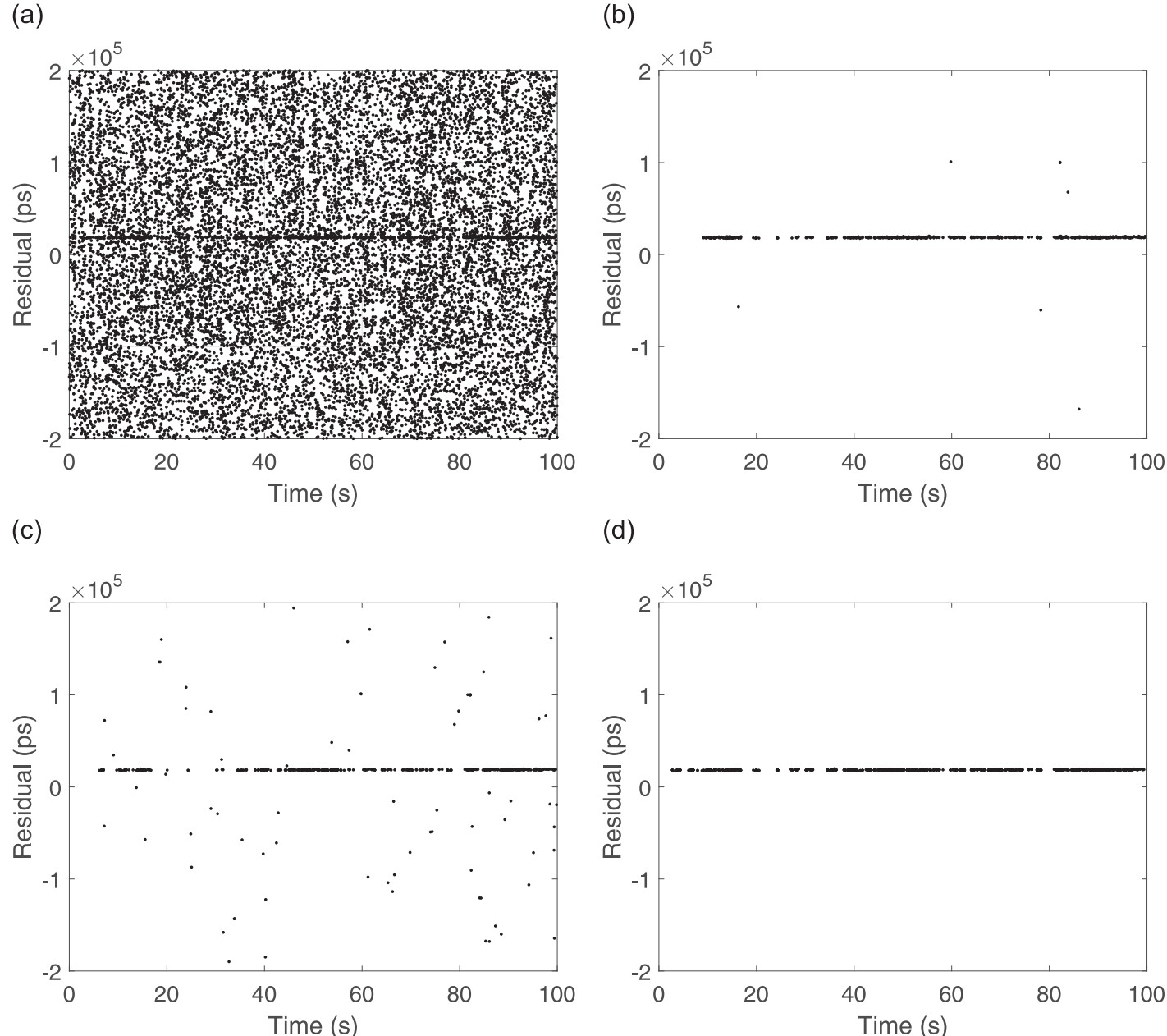

**Fig 10. Etalon2 observations and algorithmic processing results.** (a) Etalon2 raw observation data, (b) Two Graz, (c) Graz+polynomial, (d) New Algorithm.

from three satellites for 100s is processed, and the average running time of each algorithm is calculated in Fig 13.

As Fig 13 shows, the computation time of the algorithm proposed in this paper is shorter than that of the two traditional two-filter algorithms, and the algorithm processing speed is faster while ensuring measurement accuracy.

## Discussion

The results of our study demonstrate the effectiveness of the proposed filtering algorithm in addressing the limitations of traditional single-stage Graz filtering and two-stage filtering

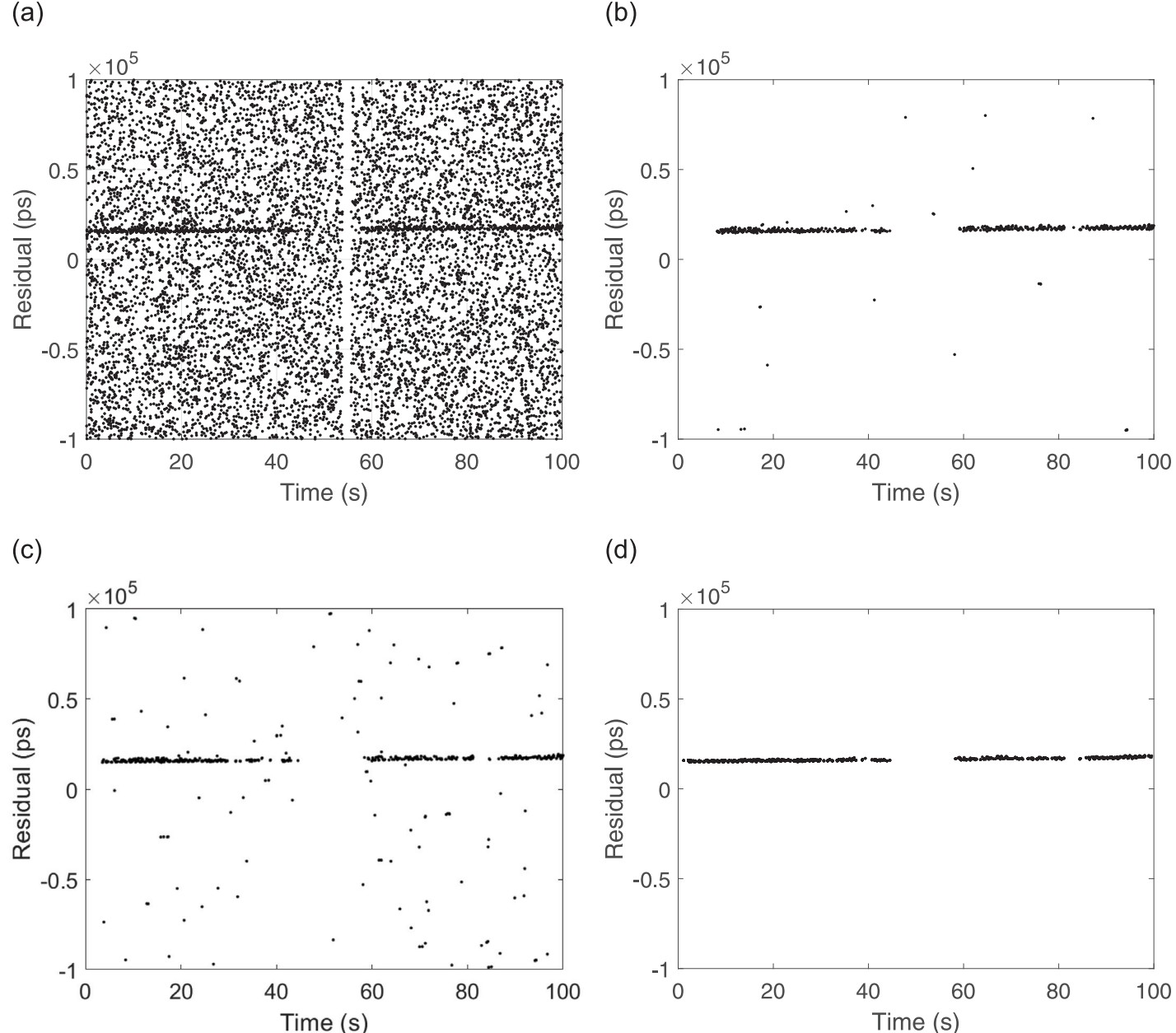

**Fig 11. Lares2 observations and algorithmic processing results.** (a) Lares2 raw observation data, (b) Two Graz, (c) Graz+polynomial, (d) New Algorithm.

algorithms. In particular, our algorithm significantly improves the identification of valid signals from medium- and high-orbit satellites, which has long been a challenge in the real-time processing of satellite laser ranging (SLR) data.

## Interpretation of findings

Compared with the single-stage Graz filtering algorithm, which struggled to identify valid signals in high-orbit satellites with a recognition rate of less than 1%, our algorithm achieved a substantial improvement. By employing multiple filtering stages, the threshold conditions of

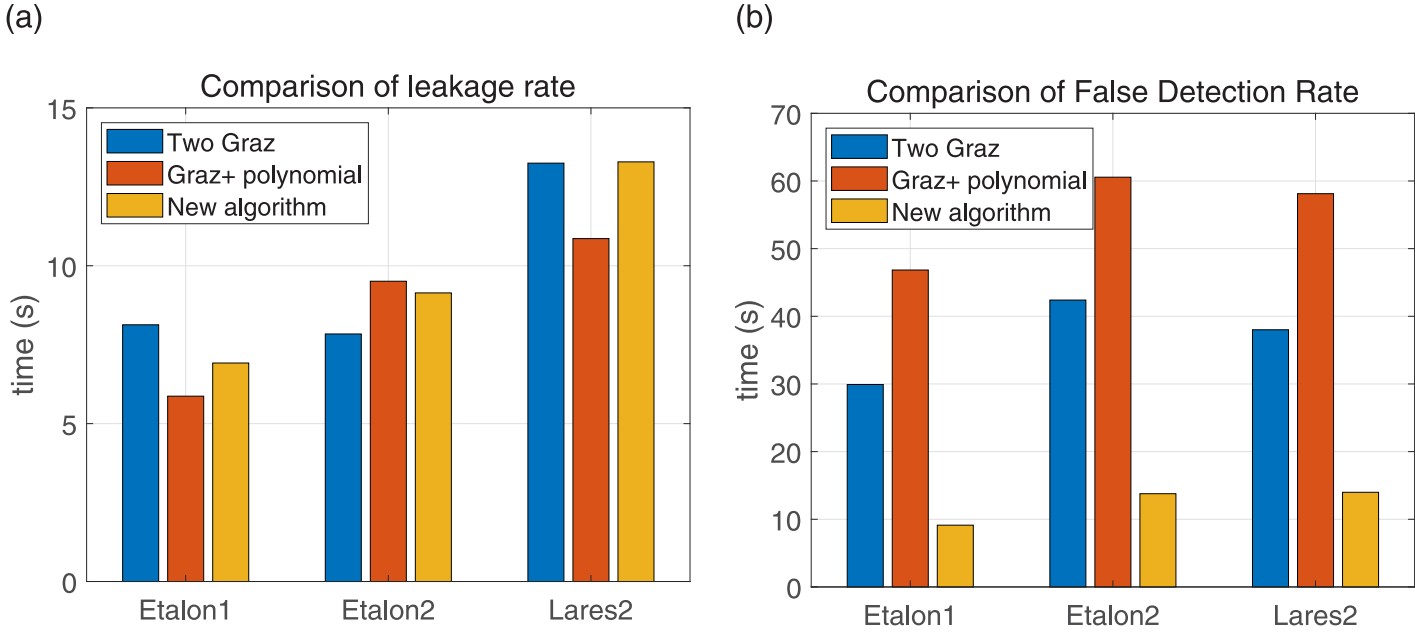

**Fig 12. Algorithm recognition result diagram.** (a) Comparison of leakage rate, (b) Comparison of false detection rate.

the initial Graz filtering can be relaxed to maximize the identification of valid signals. Through a second stage of histogram-based filtering combined with polynomial fitting, noise was effectively removed, reducing the false detection rate to below 15%. Additionally, this approach enhances signal recognition efficiency in low-echo-rate environments. This is a notable advancement in reducing missed-signal rates and ensuring more reliable data processing during real-time satellite tracking.

The strength Our method lies in its ability to accurately filter out noise while retaining weak signals, which are typically lost in traditional algorithms. For example, in the case of high-orbit satellites, the proposed algorithm identifies signals earlier than the two-stage Graz algorithm, there by reducing the signal identification delay. This early identification is crucial for real-time adjustments in satellite tracking and laser positioning, as it allow operators to align the laser beam more accurately during observations.

**Table 3. Echo signal identification results.**

| Objectives | Etalon1 | Etalon2 | Lares2 |
|---|---|---|---|
| manual screening method | 664 | 485 | 568 |
| Two Graz recognition points | 849 | 743 | 755 |
| Two Graz Fr[%] | 29.92 | 42.4 | 38.01 |
| Two Graz Lr[%] | 8.13 | 7.84 | 13.25 |
| Graz+ polynomial recognition points | 1125 | 1009 | 1077 |
| Graz+ polynomial Fr[%] | 46.84 | 60.56 | 58.12 |
| Graz+ polynomial Lr[%] | 5.84 | 9.51 | 10.86 |
| New algorithm recognition points | 679 | 525 | 572 |
| New algorithm Fr[%] | 9.13 | 13.78 | 13.99 |
| New algorithm Lr[%] | 6.92 | 9.14 | 13.29 |

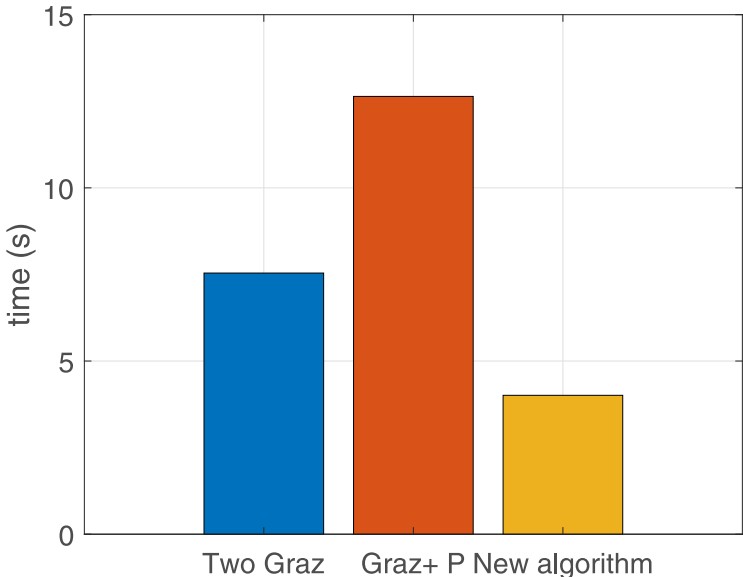

**Fig 13. Average computation time for each algorithm.**

## Basis for parameter settings

Criteria for Determining Optimal Parameter Values in the Filtering Algorithm:

1. Accuracy of Signal Identification(Minimizing False and Missed Detections). The primary criterion for parameter optimization was to achieve a low false detection rate (Fr) and a low missed detection rate (Lr). Parameters such as comparison threshold ($\delta$), and minimum valid points ($\theta$) were iteratively adjusted and tested on real-world datasets with varying signal-to-noise ratios. The goal was to balance these two metrics, ensuring accurate identification of weak and valid signals while minimizing noise recognition.

2. Real-Time Processing Efficiency. Ensuring real-time applicability required parameters that minimized computational load while maintaining accuracy. Grid sizes (e.g., large square grid width and height, small square grid step size) and sliding time window duration were tuned to optimize processing time without sacrificing the algorithm's effectiveness. Benchmarks were set based on processing speeds faster than traditional two-stage Graz filtering algorithms.

By combining theoretical insights, real-world data analysis, and iterative tuning, the optimal parameter values were systematically identified to maximize the filtering algorithm's accuracy, robustness, and real-time applicability.

## Comparison with existing studies

Our approach builds on and extends the capabilities of previous filtering techniques. For instance, while polynomial fitting and iterative filtering methods are capable of enhancing signal detection, they often result in increased false detection rates and delays in real-time recognition, particularly for high-orbit satellites. In contrast, our algorithm builds on the single-stage Graz filtering by incorporating a secondary processing step, integrating polynomial fitting within a histogram-based framework to strike a balance between minimizing false positives and ensuring timely signal recognition.

Compared to the Graz + polynomial recognition algorithm, which addresses some of the limitations of traditional filtering but still suffers from high false detection rates, our algorithm demonstrates superior performance in terms of both detection speed and accuracy. The inclusion of least-squares fitting allows our method to more effectively model signal behavior, further reducing errors during the filtering process.

## Challenges and solutions

Challenge 1: High Noise Levels in Echo Signals. Real-world data often contain significant noise due to background sky interference, system noise, and weak echo signals, especially for high-orbit satellites and space debris. These high noise levels can result in incorrect identification of noise as valid signals. Solution: The proposed algorithm incorporates a secondary filtering mechanism and polynomial fitting to reduce false detection. By leveraging the histogram-based secondary filtering approach and fitting residual maps, the algorithm effectively identifies valid signals even under noisy conditions.

Challenge 2: Weak Echo Signals from High-Orbit Satellites. Signals from medium- and high-orbit satellites are often weak, making them difficult to distinguish from noise. Traditional methods, such as the Graz algorithm, fail to identify these signals due to their low recognition rate for distant satellites. Solution: The algorithm uses a multi-stage filtering strategy, including least squares fitting to predict signal trajectories. This predictive model helps retain weak signals that might otherwise be excluded by threshold-based methods.

Challenge 3: Large Volume of Data. Real-world data involve processing a massive volume of observations, such as 100,000 data points per 100 seconds, which can significantly increase computation time and make real-time processing challenging. Solution: The algorithm was optimized for efficiency by employing a grid-based filtering mechanism that reduces computational overhead. Additionally, the histogram-based approach partitions data for parallel processing, ensuring faster computation without compromising accuracy. Traditional software displays all raw data and marks valid signals, whereas this study only displays valid signals identified by the algorithm, significantly improving the processing speed.

Challenge 4: Dynamic signals change rapidly.1. Sudden Signal Weakening: In scenarios where signal strength decreases abruptly, such as when observing high-orbit satellites or space debris, the algorithm mitigates the impact of noise by relaxing the thresholds ($\delta$) for weak signal identification. The predictive line derived from historical valid signals further aids in recognizing weak signals that might otherwise be missed.2. Sudden Signal Intensification: When the signal intensity unexpectedly increase s (e.g., due to improved atmospheric conditions), the processing of strong signal echoes resembles that of low-orbit satellites. This algorithm builds on the Graz filtering algorithm with a second filtering stage. Since the Graz algorithm achieves nearly 100% recognition accuracy for low-orbit satellite signals, the proposed algorithm is theoretically highly effective under such conditions. Experimental results further validate this performance.3. Adaptability to Changes in Signal Frequency: The algorithm operates independently of signal frequency by processing the time-series data sequentially. Adjustments to the time window and grid settings ensure compatibility with varying signal densities, allowing the algorithm to adapt to fluctuations in the frequency of detected echoes.

Challenge 5: Parameter Optimization. Setting optimal parameters (e.g., thresholds, window size, grid dimensions) required careful tuning to balance false detection and missed detection rates. Solution: Parameters were initially set based on values from previous studies and were continuously adjusted during testing with real data to ensure robust performance across various scenarios.

The challenges encountered during real-world data testing primarily revolved around noise, weak signals, computational efficiency, and signal discontinuity. By incorporating adaptive filtering techniques, predictive modeling, and grid-based data partitioning, the proposed algorithm addressed these challenges effectively, achieving high accuracy and robustness in identifying weak echo signals in real-time.

## Limitations

Despite these improvements, our method is not without its limitations. The adaptive nature of the threshold settings is crucial for balancing signal detection and noise exclusion. However, in cases where echo signals are extremely weak, even our relaxed threshold settings may result in missed detections. In Addition, while the computational efficiency of our algorithm is improved compared to traditional two-stage filtering methods, further optimization is required to handle larger datasets or more complex observation scenarios.

Another limitation of the proposed algorithm is its lack of strong adaptability. Threshold setting relies on the operator's experience and cannot be automatically adjusted based on the actual signal echo rate.

The setting of parameters such as grid size, sliding window duration, and comparison thresholds often relies on the operator's experience. This dependency introduces variability in algorithm performance, as inexperienced operators may struggle to select optimal values. Threshold settings significantly impact the algorithm's effectiveness: overly low thresholds may result in excessive noise being misidentified as valid signals. In such cases, when the laser fails to hit the target satellite, the resulting echoes, which are purely noise, could be misinterpreted as valid data. Conversely, if the threshold is set too high, the already weak echoes from medium- and high-orbit satellites may be filtered out. This could lead operators to mistakenly believe the laser missed the satellite, prompting unnecessary adjustments to the laser position and hindering observations.

The lack of threshold adaptability and reliance on operator expertise can increase false positives or missed detections, limit the algorithm's scalability, and reduce its level of automation, thereby compromising performance in diverse environments.

## Future research directions

There are several areas where the current algorithm can be further refined. The implementation of adaptive threshold mechanisms based on real-time feedback from the signal environment could enhance the flexibility of the algorithm, allowing it to adjust more effectively to varying noise levels and signal strengths. Incorporating machine learning techniques into the filtering process can also be explored as a way to improve signal classification and reduce false detections.

## Conclusion

This paper presents a novel and efficient real-time signal processing algorithm for satellite laser ranging (SLR) that addresses the challenges of weak echo signals and high noise levels, particularly in medium- and high-orbit satellites. The proposed algorithm integrates a multi-stage filtering approach, building on the speed and efficiency of the Graz filtering algorithm. By incorporating a histogram-based secondary filtering technique and applying least squares fitting to strong echo signals, the algorithm effectively reduces the false detection rate while improving the recognition of valid signals. When applied to actual measurement data and compared with two traditional quadratic filtering algorithms, it was found that the signal leakage rates of all three algorithms were comparable, approximately 10%. However, the false

detection rate was reduced from 36.78% and 55.17% to 12.30%, and this algorithm operated faster. The proposed filtering algorithm offers significant advancement in the field of satellite laser-ranging by improving both the accuracy and efficiency of signal detection. Its ability to handle weak signals and reduce false detection rates makes it a valuable tool for real-time satellite tracking and future research on space debris detection.

## Supporting information

**S1 File. The file 'Code.docx' contains the proposed algorithm and the comparison algorithms, along with detailed comments.**
(DOCX)

**S2 File. This file contains the observational data of etalon1.**
(TXT)

**S3 File. This file contains the calculated values of etalon1.**
(TXT)

**S4 File. This file contains the observational data of etalon2.**
(TXT)

**S5 File. This file contains the calculated values of etalon2.**
(TXT)

**S6 File. This file contains the observational data of lares2.**
(TXT)

**S7 File. This file contains the calculated values of lares2.**
(TXT)

## Acknowledgments

We sincerely appreciate the insights and feedback from colleagues, editors, and reviewers, which have greatly contributed to improving this research.

## Author Contributions

**Conceptualization:** Jinyu Wang.

**Data curation:** Jinyu Wang, Weiming Gong.

**Formal analysis:** Jinyu Wang.

**Funding acquisition:** Wei Zhu.

**Investigation:** Jinyu Wang.

**Methodology:** Jinyu Wang.

**Resources:** Wei Zhu.

**Software:** Jinyu Wang.

**Supervision:** Wei Zhu, Weiming Gong.

**Validation:** Jinyu Wang.

**Visualization:** Jinyu Wang.

**Writing – original draft:** Jinyu Wang.

**Writing – review & editing:** Wei Zhu, Weiming Gong.

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
