## [Decision Letter · Decision Letter 0]

5 Nov 2024

PONE-D-24-45347An Efficient Real-Time Signal Processing Method for Satellite Laser RangingPLOS ONE

Dear Dr. zhu,

Thank you for submitting your manuscript to PLOS ONE. After careful consideration, we feel that it has merit but does not fully meet PLOS ONE’s publication criteria as it currently stands. Therefore, we invite you to submit a revised version of the manuscript that addresses the points raised during the review process.

We look forward to receiving your revised manuscript.

Kind regards,

Sushank Chaudhary, Ph.D

Academic Editor

PLOS ONE

Journal Requirements:

The authors thank the Institute of Seismology, China Earthquake Administration for the financial support.

 Development of high precision time interval measuring equipment 302021-16.

Please include your amended statements within your cover letter; we will change the online submission form on your behalf."

6. We are unable to open your Supporting Information file plos_latex_template.bbl, re2.bib, plos_latex_template.aux, plos_latex_template.tex. Please kindly revise as necessary and re-upload.

Additional Editor Comments :

The authors can improve the literature review in the manuscript. The authors need to mention the keyworks on different processing Method for Satellite Laser Ranging. Moreover, provide the tabular comparison of different processing methods/algorithms with the performance of the proposed method.

Reviewers' comments:

Reviewer's Responses to Questions

**Comments to the Author**

1. Is the manuscript technically sound, and do the data support the conclusions?

Reviewer #1: Yes

Reviewer #2: Yes

2. Has the statistical analysis been performed appropriately and rigorously? 

Reviewer #1: Yes

Reviewer #2: Yes

3. Have the authors made all data underlying the findings in their manuscript fully available?

Reviewer #1: Yes

Reviewer #2: Yes

4. Is the manuscript presented in an intelligible fashion and written in standard English?

Reviewer #1: Yes

Reviewer #2: Yes

5. Review Comments to the Author

Reviewer #1: the authors of manuscript "An Efficient Real-Time Signal Processing Method for Satellite Laser Ranging" provided a methor to a kind of real-time SLR data processing method, and solved the challenges brought by the weak echo signal and high noise level of the high-orbit satellite clock. The research method is highly innovative, the result improvement is significant, and the research content conforms to the scope of PLOS ONE.

Reviewer #2: This authors proposed a new multi-stage filtering algorithm to improve signal processing for satellite laser ranging (SLR). The authors claim that their method addresses challenges related to weak echo signals and high noise levels, especially for medium- and high-orbit satellites. The proposed algorithm builds on the Graz filtering method and incorporates a histogram-based secondary filtering technique and least squares fitting.

The proposed algorithm shows promise in enhancing the identification of valid signals from high-orbit satellites. The authors have also clearly explained how the multiple filtering stages improve performance compared to traditional methods. This research also addresses important issues in real-time satellite tracking and space debris detection.

I recommend this paper publication after the authors address the following concerns.

1. What challenges were encountered when testing the algorithm with real-world data, and how were they addressed?

2. While the paper acknowledges limitations regarding threshold adaptability and the reliance on operator experience, it would benefit from a more in-depth exploration of how these limitations might impact practical applications in various environments.

3. What criteria were used to determine the optimal values for the parameters used in your filtering algorithm?

4. The paper makes valuable comparisons with traditional algorithms; it would be significantly enhanced by incorporating a wider range of diverse filtering techniques for evaluation.

5. How does the algorithm perform with dynamic signals that may change rapidly during observations? Is it capable of adapting to sudden changes in signal strength or frequency?

6. PLOS authors have the option to publish the peer review history of their article (what does this mean?). If published, this will include your full peer review and any attached files.

Reviewer #1: No

Reviewer #2: No

---

## [Author Response · Author response to Decision Letter 0]

18 Nov 2024

We sincerely thank the editors and reviewers for their valuable suggestions. Based on their feedback, we have made the corresponding revisions, which are detailed in the "Response to Reviewers" document.

---

## [Decision Letter · Decision Letter 1]

26 Nov 2024

An Efficient Real-Time Signal Processing Method for Satellite Laser Ranging

PONE-D-24-45347R1

Dear Dr. zhu,

We’re pleased to inform you that your manuscript has been judged scientifically suitable for publication and will be formally accepted for publication once it meets all outstanding technical requirements.

Kind regards,

Sushank Chaudhary, Ph.D

Academic Editor

PLOS ONE

Additional Editor Comments (optional):

Reviewers' comments:

Reviewer's Responses to Questions

**Comments to the Author**

1. If the authors have adequately addressed your comments raised in a previous round of review and you feel that this manuscript is now acceptable for publication, you may indicate that here to bypass the “Comments to the Author” section, enter your conflict of interest statement in the “Confidential to Editor” section, and submit your "Accept" recommendation.

Reviewer #1: All comments have been addressed

Reviewer #2: All comments have been addressed

2. Is the manuscript technically sound, and do the data support the conclusions?

Reviewer #1: Yes

Reviewer #2: Yes

3. Has the statistical analysis been performed appropriately and rigorously? 

Reviewer #1: Yes

Reviewer #2: Yes

4. Have the authors made all data underlying the findings in their manuscript fully available?

Reviewer #1: Yes

Reviewer #2: Yes

5. Is the manuscript presented in an intelligible fashion and written in standard English?

Reviewer #1: Yes

Reviewer #2: Yes

6. Review Comments to the Author

Reviewer #1: (No Response)

Reviewer #2: The authors have addressed all the questions and concerns raised during the peer-review process. They have made significant revisions, which have substantially improved the clarity and robustness of the paper. The proposed multi-stage filtering algorithm for satellite laser ranging (SLR) is a promising and innovative solution. Given the thorough revisions and the clear clarification of key points, I recommend that the manuscript be accepted for publication.

7. PLOS authors have the option to publish the peer review history of their article (what does this mean?). If published, this will include your full peer review and any attached files.

Reviewer #1: No

Reviewer #2: No

---

## [Editor Report · Acceptance letter]

9 Dec 2024

PONE-D-24-45347R1 

PLOS ONE

Dear Dr. Zhu, 

I'm pleased to inform you that your manuscript has been deemed suitable for publication in PLOS ONE. Congratulations! Your manuscript is now being handed over to our production team.

Kind regards, 

on behalf of

Prof. Sushank Chaudhary 

Academic Editor

PLOS ONE